# Influence of the Thermal Insulation Type and Thickness on the Structure Mechanical Response Under Fire Conditions

**Katarzyna Kubicka** , **Urszula Pawlak \* and Urszula Radoń**

Faculty of Civil Engineering and Architecture, Kielce University of Technology, 25-314 Kielce, Poland
\* Correspondence: u.pawlak@tu.kielce.pl; Tel.: +48-41-34-24-803

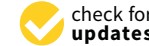

**Featured Application:** **The problems analysed in the study are extremely important from the point of view of the fire safety of steel structures. In technical approvals, manufacturers of fire-proofing materials usually provide tables in which the necessary insulation thicknesses are listed, depending on the type of material used, heating conditions, section factor, and the required fire resistance class. The manufactures of fireproofing materials recommend that insulation of a constant thickness be used for all elements of the truss. This approach seems highly uneconomical. An alternative solution has been proposed in the paper. This proposal determines the optimal necessary insulation thickness for individual bars of the truss, based on additional static-strength calculations. This article describes the original approach to modelling protected steel structures against fire. The advantage of the proposed solution is a more optimal (economical) method of fire insulation for steel lattice girders.**

**Abstract:** The concept of fire safety covers an extremely vast scope of issues. To ensure an adequate fire safety level, it is necessary to combine research and actions in several fields, such as the mathematical, physical, or numerical modelling of a fire phenomenon. Another problem is to design different types of fire protection, including alarm systems, sprinkler systems, and also roads and evacuation systems, in a manner that ensures maximum safety for the building's users. A vital issue is the analysis of the static-strength response of the structure under fire conditions. This study, concerned with such analyses, is limited to steel truss structures. In technical approvals, manufacturers of fire-proofing materials do not account for the character of the performance of individual structural members. The components in compression need thicker insulation than those in tension. This phenomenon is related to the fact that under fire conditions, the flexural buckling coefficient in compressed members is abruptly reduced with an increase in temperature. In turn, this increase in temperature leads to a fast reduction in resistance. In addition, members in tension have much higher resistance than those in compression in the basic design situation, i.e., at the instant of t = 0 min. Consequently, even a considerable decrease in the resistance of tension members is not as dangerous as that of compression members. Therefore, due to the nature of the performance of individual elements, fire-proofing insulation of every steel structure should be computationally verified. Additionally, in this paper, the influence of the type of fire insulation on the mechanical response of the structure was investigated. Calculations were carried out for different types of sprayed-on insulation, and also for contour and box insulation panels. The graphs show the behaviour of the elastic modulus, the yield point, and the resistance of the elements in the successive minutes of the fire for the different methods of fire protection used. The best results were obtained for vermiculite and gypsum spray.

**Keywords:** fire insulation; fire analysis; steel structures; sprayed-on insulation; fire-proofing boards

---

## 1. Introduction

Steel offers many advantageous properties, including high strength and high resistance. Those properties make steel an almost ideal structural material. However, under fire conditions, steel properties deteriorate with exposure to elevated temperatures. This deterioriation leads to a decrease in the resistance of individual structural elements, which may result in a failure or collapse of the structure.

In a fire situation, the main priority is to protect human life. Next, the structure should be protected. These two goals are closely related. Improper protection of the building often prevents efficient evacuation of people, which may result in a calamity. Therefore, when a structure is designed, it must be provided with an appropriate fire resistance class to give people a long enough time to escape. All measures should be taken to reduce the risk of a structural collapse [1–3]. Then, it is necessary to ensure that after the fire, the structure can still be brought back into use with minimum necessary repairs.

One of the most catastrophic structure disasters in history was the collapse of the Twin Towers of New York City's World Trade Center. Media coverage created a public belief that the towers collapsed due to the impact of an aircraft. Numerous studies and expert opinions produced after the disaster, however, show that this impact was only an indirect cause. The impact of the aircraft caused a fire, which became a direct cause of the destruction of the towers [4–9]. Fires in the tall and high-rise buildings had happened before, but the September 11th disaster was the only one in which the steel skyscraper collapsed completely.

Some technical solutions that work perfectly well under normal conditions become a major disadvantage during a fire. In extreme situations, steel structures exposed to fire can disintegrate very quickly. This happened in Lubań, Poland, in July 2012, as a result of a fire in the production facility. The steel building, designed and made from thin-walled steel elements with a thickness not greater than 3 mm, collapsed in just 17 min [10]. The use of cold formed structural elements has become more common in the construction industry during the past decades due to the various advantages cold formed structures offer, such as weight reduction of the structural system. The thin-walled sections differ from hot-rolled sections in terms of failure modes. In addition to local and global buckling, distortional buckling is also possible. The slenderness of steel members is a major factor in determining their failure mode at elevated temperatures. Short, stocky elements typically have a slenderness ratio less than 40, while for long, slender elements this value is greater than 120. The stocky columns exhibited local buckling as their failure mode, while longer, more slender columns displayed global buckling accompanied by limited local buckling. Problems related to the description of failure modes have been the subject of discussion in many works. Articles that deserve special attention include those in [11–13].

The fire safety of steel structures does not only refer to buildings. Fires of bridges do not happen often, but if they occur, they produce serious consequences. The effects associated with a bridge being put out of operation are felt by a significant number of residents of the area and often the entire economy of the region. That was the case after the fire of the Łazienkowski bridge in Warsaw in February 2015. An average of 95,000 vehicles travelled daily over the bridge (about 18% of vehicular traffic on all Warsaw bridges). The initial expert report of the bridge's technical condition showed degradation and a loss of carrying capacity in, among other elements, the steel structure of the bridge, resulting from elevated temperatures. [14]. On February 20, 2015, the city authorities decided to dismantle the damaged steel structure of the bridge and mount a new one on the surviving pillars. The team from the Warsaw University of Technology, headed by Professor Zobel, conducted a detailed post fire inspection, which included materials tests, verification of the construction geometry and Finite Element Method (FEM) strength analysis [15]. Experts found numerous additional defects, including dilatation damage, failure of the steel structure of the access walkway, damages to the road surface, protective barriers, paint coatings, and utilities. It was concluded that the main cause of damage was thermal shrinkage of steel. Due to various boundary conditions for individual elements

of the bridge, a number of residual deformations occurred as the structure cooled. That resulted in selected elements being put out of operation. Finally, a decision was made to replace the steel structure of the spans using a modified incremental launching method. The bridge was opened on October 28, 2015.

The reported cases of fires that have occurred in recent years clearly indicate the need to improve the methods and means of increasing the fire safety of structures. The concept of fire safety is an extremely extensive issue. Ensuring an appropriate level of fire safety has been implemented in research and activities in several fields, especially mathematical, physical, and numerical modeling of fire phenomena. Another issue is the design of all types of fire protection, such as types of alarm systems or sprinkler systems, and the design of roads and evacuation systems in a manner ensuring maximum safety for building users. Fire protection covers all types of structures, due to the nature of the work, as well as the material used. Currently popular composite materials, such as concrete filled steel tube (CFST) columns, papers [16–18], and polypropylene fibers [19] are used in the area of the structures themselves. In contrast, the connections of load-bearing elements are modeled with the participation of a high-performance fiber reinforced cement composite (HPFRCC) [13]. The HPFRCC high performance fiber-reinforced cementitious composite is a mixture of the correct proportions of Portland cement, fly ash, quartz sand, water, and polyvinyl alcohol (PVA) fibers. It is PVA fibers that have a decisive influence on improving the mechanical efficiency, elasticity, and durability of concrete structures [20,21] due to fire conditions. According to Li, Xu, Bao, and Cong [22], PVA fibers under the influence of high temperatures melt to form channels to alleviate internal vapor pressure and thus explosive spalling. Therefore, the mechanical properties after a fire are higher compared to those of conventional concrete. In [22], the authors performed experimental studies of four two-story and two-nave frames, two made of typical concrete (S1 and S2) and two in which the beam–column joint was made of high-performance cement composite reinforced with HPFRCC fibers (S3 and S4). The frames S2, S3, S4 were subjected to fire. The results of the tests carried out in the fire chambers showed that the application of HPFRCC increased the initial stiffness by 30%, the resistance and ultimate displacement by 6% and 3%, and the energy dissipation by 21%. In addition, only small surface flakes appeared at the HPFRCC joints, without the participation of explosive spalling on the HPFRCC. In works [23] and [24], as thermal insulation, polystyrene foam was used and in [25], a portable compressed air foam system was used (regardless of its material of construction). An extremely important problem is the analysis of the static-strength response of structures in fire conditions [26]. This paper focuses on this field and narrows it down to steel trusses. In steel structures, fire insulation is used in the form of intumescent paints or fireproof spray materials. The properties of these materials are discussed in other scientific articles [1,27,28]. In this paper, thermal vermiculite mortar, vermiculite fireproof plaster with gypsum, protection in the form of vermiculite fire protection boards with cement, as contour insulation (for all types of cross-sections), and as box insulation (only for I-sections), were adopted as thermal insulation. In technical approvals of their products, the manufacturers of fire insulation recommend the adoption of a constant insulation thickness for all elements of the analyzed structure. However, such a solution is scarce (unprofitable). In this paper, we propose to carry out additional static-strength calculations in order to determine the optimal, required insulation thickness for each type of member truss.

## 2. Thermal and Strength Analysis of Compression and Tension Steel Elements in Fire

### 2.1. Thermal Analysis

Designing a structure for an accidental situation, i.e., a fire, is based on an analysis of thermal interactions. With an increase in the temperature of fire gases $\theta_g$, the temperature of the steel element increases $\theta_a$ and its resistance $N$ decrease. Fire engineering requirements for the assumed time $t_{fi,req}$ are met if either of two conditions is satisfied:

- $\theta_{cr} \geq \theta_a$—Check in the temperature domain;

- $N \geq E$—Check in the resistance domain.

The temperature domain check involves a comparison of the temperature that the steel element achieves ($\theta_a$) after a specified fire duration with its critical temperature $\theta_{cr}$. The disadvantage of this approach lies in some limitations concerning the critical temperature determination and in the fact that this approach cannot be applied to some elements. Consequently, resistance domain checks are more popular. This process involves comparing the resistance of the element ($N$) with the effect of actions ($E$). The latter is the value of forces generated in the structure due to an external load. To describe the temperature of the fire gases, a standard fire curve was adopted:

$$\theta_g = 345 \, \log_{10}\left(8 \, t_{fi} + 1\right) + 20, \, °\text{C}. \tag{1}$$

When the temperature of fire gases $\theta_g$ is known, it is possible to determine the temperature that individual elements of the structure will reach.

### 2.1.1. Element Temperature Increment without Thermal Insulation

For unprotected steel elements, the temperature increment can be estimated according to the formula:

$$\Delta\theta_a = k_{sh} \frac{A_m/V}{c_a \rho_a} \dot{h}_{net,d} \Delta t_{fi}, \tag{2}$$

where:

$k_{sh}$—correction factor for the shadow effect (dependent on the section factor);

$A_m/V$—section factor for unprotected steel members according to Table 4.2 PN-EN 1993-1-2, where $A_m$ is the surface area per unit length, and $V$ is the member volume per unit length;

$c_a$—specific heat of steel;

$\dot{h}_{net,d}$—net heat flux per unit area according to [29];

$\Delta t_{fi}$—time interval (not greater than 5 s);

$\rho_a$—the unit mass of steel.

### 2.1.2. Insulated Element Temperature Increment

When temperature distribution in the section is uniform, the temperature increment $\Delta\theta_a$ of the insulated steel member in the time interval $\Delta t$ is given by the following formula [30]:

$$\Delta\theta_a = \frac{\lambda_p A_p/V\left(\theta_g - \theta_a\right)}{d_p c_a \rho_a \left(1 + \phi/3\right)} \Delta t_{fi} - \left(e^{\phi/10} - 1\right)\Delta\theta_g, \tag{3}$$

and

$$\phi = \frac{c_p \rho_p}{c_a \rho_a} d_p \, A_p/V \tag{4}$$

where:

$A_p/V$—section factor for steel members insulated by fire protection material, determined from Table 4.3 [30];

$c_a$—temperature-dependent specific heat of steel;

$c_p$—temperature-independent specific heat of fire protection material;

$d_p$—thickness of fire protection material;

$\Delta t_{fi}$—time interval (not greater than 30 s);

$\theta_a$—temperature of steel members;

$\theta_g$—temperature of fire gases;

$\Delta\theta_g$—temperature increment of fire gases in the time interval $\Delta t_{fi}$;

$\lambda_p$—thermal conductivity of the fire protection material;

$\rho_a$—unit mass of steel;

$\rho_p$—unit mass of the fire protection material.

## 2.2. Design Resistance of Tension and Compression Steel Members in Fire

### 2.2.1. Tension Elements

The design resistance of a tension member with a uniform temperature across its cross-section is given by formula [30]:

$$N_{fi,\theta,Rd} = k_{y,\theta_a} N_{Rd} \left[ \gamma_{M,0} / \gamma_{M,fi} \right] \tag{5}$$

where:

$N_{Rd}$—design resistance of the gross cross-section $N_{pl,Rd}$ for normal temperature design according to [29];

$k_{y,\theta_a}$—reduction factor for the yield strength of steel at temperature $\theta_a$ reached by the element in the fire at time $t_{fi}$, according to Table 1;

**Table 1.** Reduction factors for yield strength and modulus of elasticity [30].

| Steel Temperature $\theta_a$ | Reduction Factors at $\theta_a$ Temperature Relative to $f_y$ or $Y$ at 20 °C | |
| --- | --- | --- |
| | **Reduction Factor of the Effective Yield Strength (relative to $f_y$)** $k_{y,\theta_a} = f_{y,\theta_a}/f_y$ | **Reduction Factor of the Modulus Of Linear Elasticity (relative to $Y$)** $K_{Y,\theta_a} = Y_{\theta a}/Y$ |
| 20 °C | 1.000 | 1.000 |
| 100 °C | 1.000 | 1.000 |
| 200 °C | 1.000 | 0.900 |
| 300 °C | 1.000 | 0.800 |
| 400 °C | 1.000 | 0.700 |
| 500 °C | 0.780 | 0.600 |
| 600 °C | 0.470 | 0.310 |
| 700 °C | 0.230 | 0.130 |
| 800 °C | 0.110 | 0.090 |
| 900 °C | 0.060 | 0.0675 |
| 1000 °C | 0.040 | 0.0450 |
| 1100 °C | 0.020 | 0.0225 |
| 1200 °C | 0.000 | 0.0000 |

$\gamma_{M,0}$ —partial factor used when checking the load capacity of the cross-section;

$\gamma_{M,fi}$ —partial factor for the relevant material property for the fire situation.

### 2.2.2. Compression Elements

For compression members, with sections having Class 1, Class 2 or Class 3 proportions, with a uniform temperature across its cross-section, the design buckling resistance in the fire at time is expressed by formula [30]:

$$N_{b,fi,\theta,Rd} = \chi_{fi} A k_{y,\theta_a} \, f_y \, / \gamma_{M,fi} \tag{6}$$

where:

$\gamma_{M,fi}$—partial factor for material properties in a fire situation;

$k_{y,\theta_a}$ —reduction factor for the yield strength of steel at temperature $\theta_a$ reached in the fire at time $t_{fi}$, according to Table 1;

$A$—cross-sectional area;

$f_y$ —yield strength;

$\chi_{fi}$ —flexural buckling coefficient in the design fire situation calculated according to the formula:

$$\chi_{fi} = \frac{1}{\phi_{\theta_a} + \sqrt{\phi_{\theta_a}^2 - \overline{\lambda_{\theta_a}}^2}} \tag{7}$$

and

$$\phi_{\theta_a} = \frac{1}{2}\left[1 + \overline{\alpha\lambda_{\theta_a}} + \overline{\lambda_{\theta_a}}^2\right] \tag{8}$$

and

$$\alpha = 0.65\sqrt{235/f_y} \tag{9}$$

Non-dimensional slenderness $\overline{\lambda_\theta}$ at temperature $\theta_a$ is given by the formula:

$$\overline{\lambda_{\theta_a}} = \overline{\lambda}\left[k_{y,\theta_a}/k_{Y,\theta_a}\right]^{0.5} \tag{10}$$

where:

$\overline{\lambda}$—non-dimensional slenderness in the basic design situation,

$k_{y,\theta_a}$, $k_{Y,\theta_a}$—respectively, the reduction factor of the yield point and the modulus of linear elasticity of steel at temperature $\theta_a$, according to Table 1.

## 3. Result and Discussion

The structure shown in Figure 1a is a steel, industrial hall with lattice girders pinned jointed to prefabricated reinforced concrete columns. The bracing system commonly used in this type of building makes it possible to model lattice girds as flat trusses (Figure 1b). This approach is widely used in design practice. The connection of a steel lattice girder and a column is implemented as pinned. For this reason, it was assumed that the truss is simply supported. The system was loaded only with concentrated forces applied to the nodes of the top chord. In Figure 1b, the compression elements are marked in red and the tension members are marked in green.

This structure was analysed with respect to basic and accidental (i.e., fire design) situations. It was assumed that all structural elements are made of S275 steel and heated from all sides. The top and bottom chords were made from I-sections, whereas bracing members were made from square tubing. The selected profiles and the effect of actions, i.e., forces arising in individual elements of the structure, are shown in Table 2.

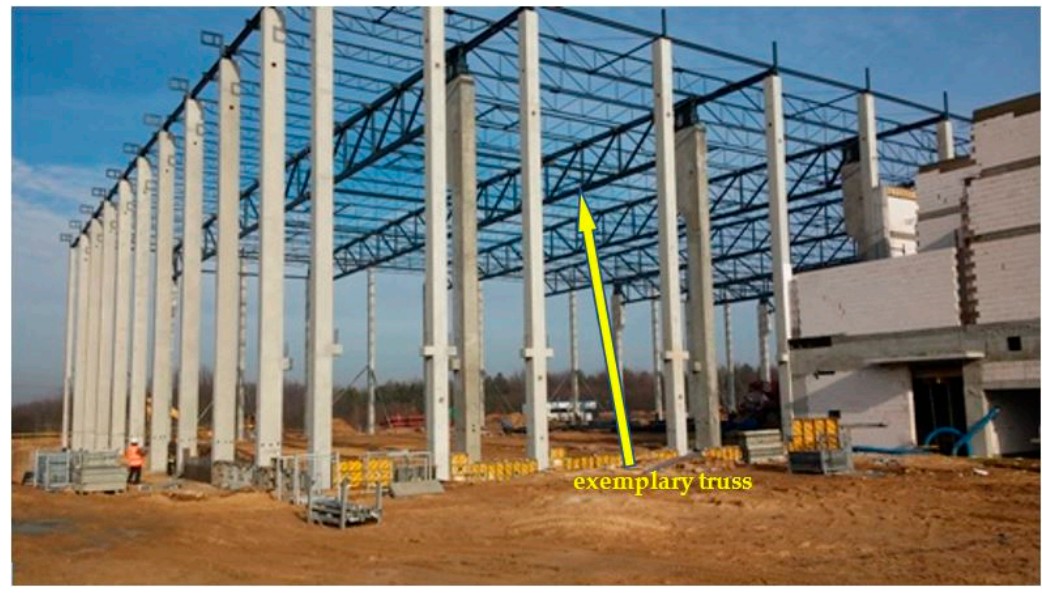

**Figure 1.** (**a**) Example of an industrial hall with steel lattice girders [31]. (**b**) Static scheme of the truss model.

**Table 2.** Truss sections and the effect of actions.

|  | Section | Effect of Actions |
|---|---|---|
| **Bottom chord** | I100 | $N_1 = N_6 = 0$ kN; $N_2 = N_3 = N_4 = N_5 = 53$ kN |
| **Top chord** | IPE 140 | $N_7 = N_8 = N_{11} = N_{12} = -38$ kN; $N_9 = N_{10} = -54$ kN |
| **Bracing** | RHS50x50x5 | $N_{13} = N_{19} = -36$ kN; $N_{14} = N_{16} = N_{18} = -12$ kN; $N_{15} = N_{17} = 0$ kN <br> $N_{20} = N_{25} = 46$ kN; $N_{21} = N_{24} = -21$ kN; $N_{22} = N_{23} = 1$ kN |

In the static analysis, by means of FEM, we used a flat truss element, which is a simple two-node element with two degrees of freedom in each node. The shape functions of the displacement field are linear. The displacement field of an element contains two translational components: horizontal u and vertical v. The strain tensor is reduced to one non-zero component of Green's strain tensor $\varepsilon_{11}$, which characterizes the elongation of the bar. Other components of the strain's tensor are equal to zero. The stress tensor is represented by component $\sigma_{11}$ of Pioli-Kirchhoff's second stress tensor. The detailed mathematical formulations of the stress-strain relationship for steel at elevated temperatures are given in [30]. In Figure 2, this relationship is shown for steel S275.

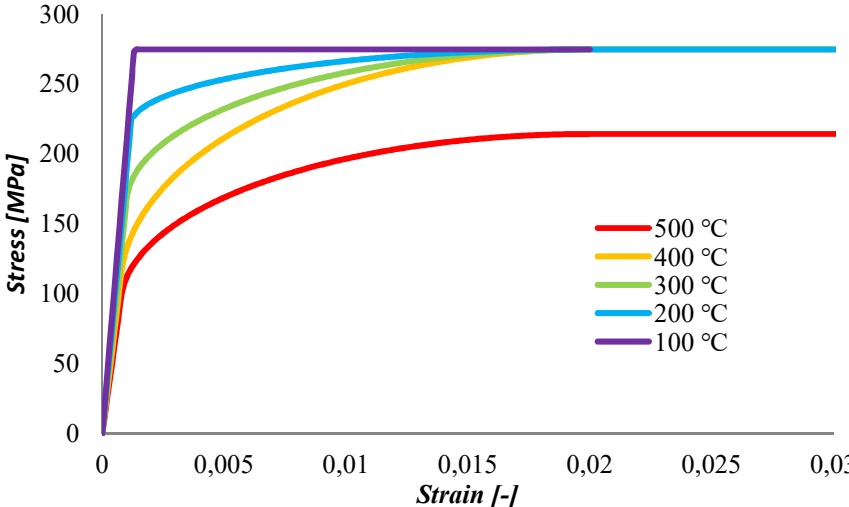

**Figure 2.** Stress–strain relationship for steel S275.

The starting point of a full fire analysis is to define fire scenarios. It is assumed that the structure is in the fully developed fire phase, so the standard fire curve (Equation (1)) is adopted. For unprotected and insulated steel members, the temperature increment was estimated in accordance with the formulas in Equations (2) and (3). The temperature field is uniform along the member length and along the cross-section height. Next, the mechanical properties that influence the resistance of tension and compression members are calculated. The most important mechanical properties of steel elements are yield strength and Young's modulus of elasticity. A decrease in the values of mechanical parameters leads directly to a reduction in the resistance of tension and compression members, as per Equations (5) and (6), which finally results in the limit state being exceeded. Thermal load is brought into the nodes as an additional load, in accordance with the incremental FEM method. For thermal load defined in this way, the stiffness matrix of the truss element is determined for Young's modulus as a function of temperature.

The fire analysis is not intended to determine the value of the limit load but that of fire resistance, i.e., to specify the time of the structure's failure. In this particular case, static loads are constant, whereas additional axial forces can be created in the truss elements under thermal load. Due to the fact that the structure is statically determinate, during the whole fire analysis, internal forces remain the same as in the basic situation. The computations, aimed at finding axial forces, strains, and displacements, are run with the MES3D program [32–34]. The computations proceeded in two steps. The first step (20 °C) produced the specification of the state of the structure in a basic design situation. In the second step, temperature increments related directly to time steps, are taken into account. The analysis was repeated every second.

The simplest and most effective way to protect steel structures in the event of a fire is to isolate them from the influence of high temperature using the following protection [35]:

- Thermally activated (reactive) agents, e.g., swelling coatings;
- Thermally passive means, e.g., spray masses and plate claddings;
- Hybrid means combining plate claddings and spray masses.

Thermal properties of insulating materials change as fire develops [28,36–38]. However, the Polish standard [29] allows adopting them as fixed values.

In this paper, fire analysis of the structure was carried out using three different types of thermal insulation with a thickness of 2 cm:

- Protection with vermiculite fireproofing mortar;
- Protection with vermiculite mortar with gypsum (high density);

- Protection in the form of vermiculite boards with cement, as contour insulation (for all types of cross-sections) and as box insulation (only for I-sections).

The characteristics of the selected fire-proofing materials are listed in Table 3.

**Table 3.** Thermal properties of selected fire-proofing materials [35].

| Fire-Proofing Insulation Material | Unit Mass $\rho_p$, kg/m3 | Thermal Conductivity $\lambda_p$, W/(mK) | Specific Heat $C_p$, J/(kgK) |
|---|---|---|---|
| vermiculite | 350 | 0,12 | 1200 |
| vermiculite (or perlite) and gypsum | 650 | 0,12 | 1100 |
| vermiculite boards (or perlite) and cement | 800 | 0,20 | 1200 |

Sprayed-on coatings are the most popular type of fireproofing insulation because of the many advantages they offer, including low thermal conductivity, low self-weight, cost-effectiveness, and simple installation [28,37].

Sprayed-on fireproofing pastes are produced in the form of dry mix, the ingredients of which are [35]: cement- or gypsum-based binder, aggregate (e.g., vermiculite, perlite, mineral fibre, or mineral (rock) wool granulate), and modifiers.

Sprayed-on pastes can be applied in wet or dry technology. Currently, the most commonly used is a dry technology, consisting of transporting a pneumatically dry mix, which is combined with water or a liquid binder at the outlet end of the sprayer. Wet technology is less frequently used, i.e., mechanical application of water-mixed mixture using pump-spray units in a manner similar to machine plastering [2,35]. The all wet process involves dirt and wetting the surroundings. Fireproof spraying is performed based on the thickness in one or several layers, which are most often 2 or 3 layers. The thickness of the insulation from the spray masses ranges from 15 to 60 mm. The fire resistance class obtained with the participation of spraying fire-retardant masses is R15–R240.

Fireproofing protection of steel structure members can be provided using the following types of boards [35]: rock wool, gypsum plasterboard, magnesium oxide-based, those that have gypsum or cement binder, those with cement and lime binders with reinforcement (most often a glass fibre one), and various types of filler materials.

Board insulations can be installed using different designs, namely contour or box configurations. In the first design, boards are fixed in such a way that the protected element cross-sectional shape is maintained. The other configuration involves the formation of the cuboid box, inside which the protected section is built-in.

Board claddings are aesthetically appealing, and they usually do not need additional finishing, but they provide a more expensive fire protection option compared with sprayed-on or intumescent coatings. Additionally, the time necessary for installation is much longer than for fireproofing coatings. That contributes to higher investment costs and extends the duration of the construction works [1]. Board protection, applied as single or multilayer cladding, allows one to achieve fire resistance class R15–R240 [35].

In accordance with Equations (2) and (3), the temperature increment in steel members also depends, among others, on the section factor. For sprayed-on contour board insulation, this factor is equal to the perimeter to the sectional area ratio. For box insulation, this factor is computed from formula [29]:

$$\frac{A_p}{V} = \frac{2(b+h)}{A_S} \qquad (11)$$

where $b$ is I-section width, $h$ is section height, and $A_s$ is the cross-sectional area of the profile.

The highest section factor is found for the I100 section, used in the bottom chord. The value is equal to 349.057 m$^{-1}$ for sprayed-on and contour board insulation. The value of box board insulation is reduced to 283.019 m$^{-1}$. The top chord sections, namely IPE140, with box insulation show an exposure factor of 259.756 m$^{-1}$ and 335.976 m$^{-1}$ for other types of insulation. With respect to the bracing elements, made from square tubing RHS50x50x5, the exposure factor equals 214.204 m$^{-1}$.

The temperatures of unprotected members, and the temperatures of all sections used in the experiment with different insulation types, are shown in Figure 3.

The results presented above indicate that as the fire develops, the temperature of the members without fire-proofing insulation becomes closer to and, over time, equal to the temperature of fire gases. In such a situation, an abrupt fall in the modulus of elasticity and yield strength occurs, which results in a substantial reduction in the section resistance, leading to the exceedance of the ultimate limit state. Hence, further analysis involved only the members with fire-proofing insulation.

The thermal analysis was followed by the examination of the effect produced by different types of insulation on a decrease in the modulus of elasticity (Figure 4) and yield strength (Figure 5). The approach proposed in Eurocode [30], namely reduction factors (Table 1), was applied.

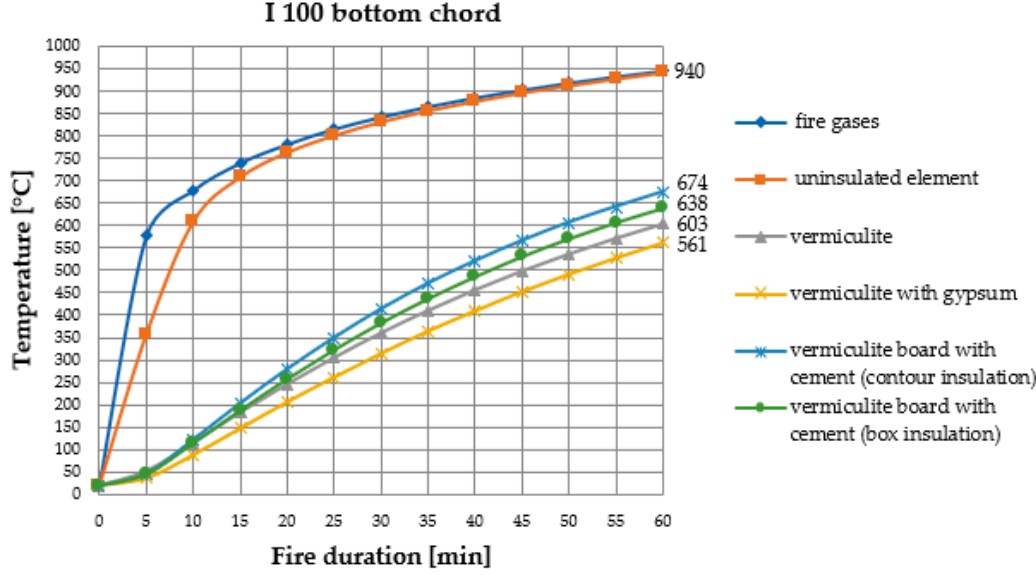

(a)

**Figure 3.** *Cont.*

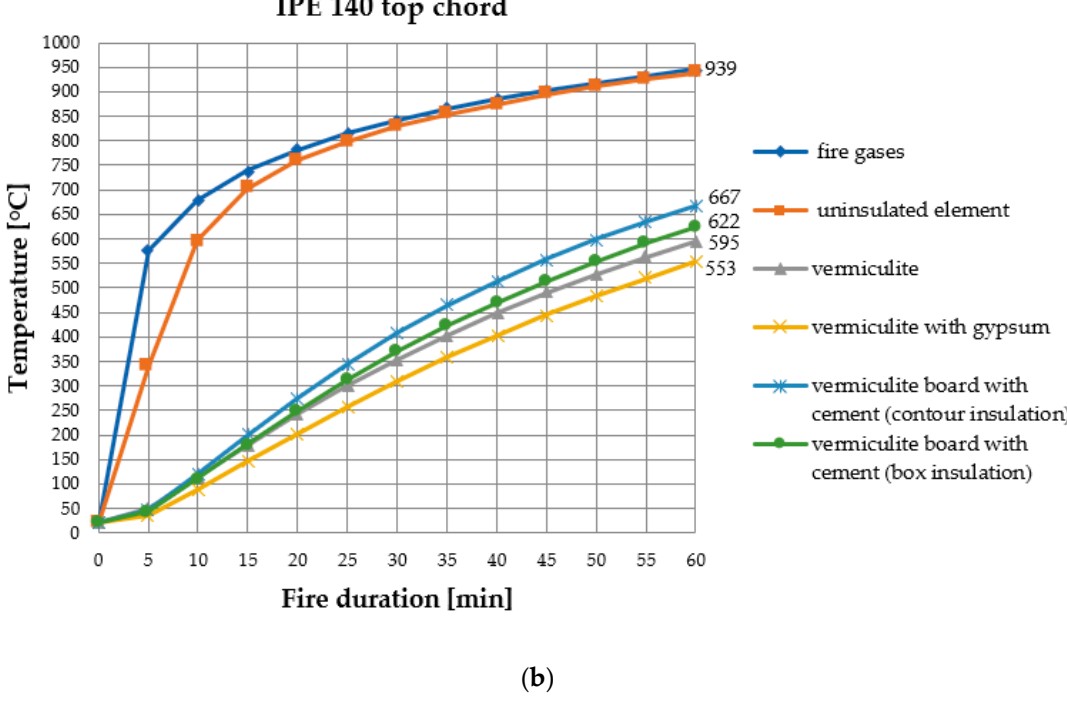

(**b**)

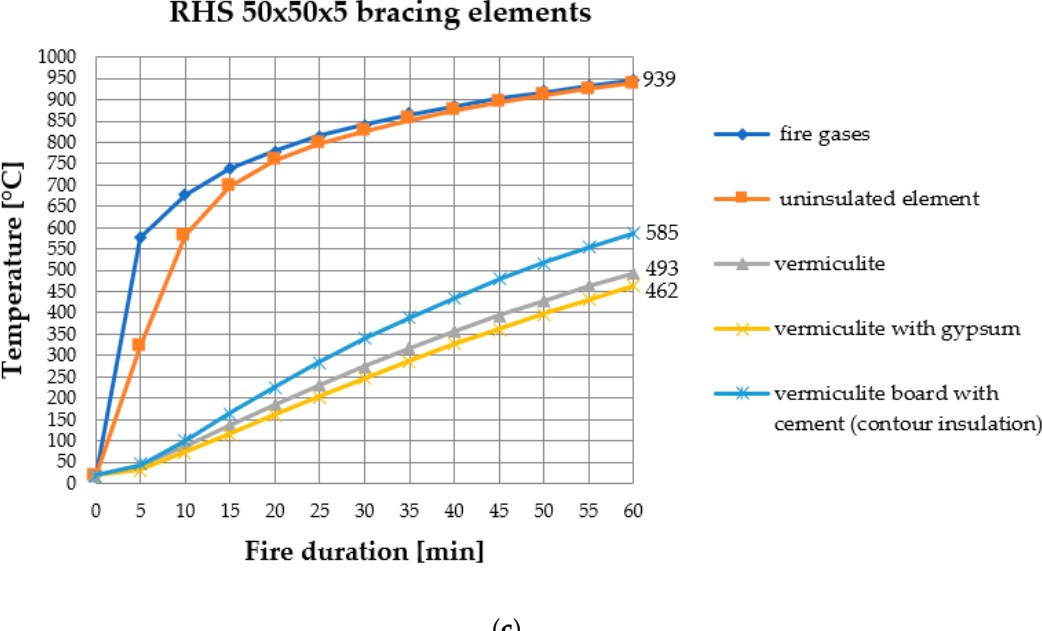

(**c**)

**Figure 3.** Temperature of steel members in the successive minutes of the fire (insulation 2 cm): (**a**) I 100 bottom chord, (**b**) IPE 140 top chord, (**c**) RHS 50 × 50 × 5 bracing elements.

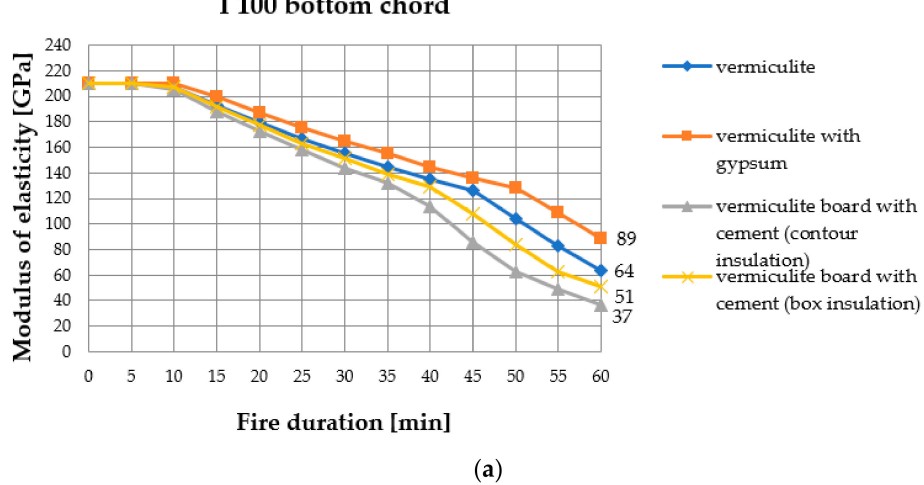

(**a**)

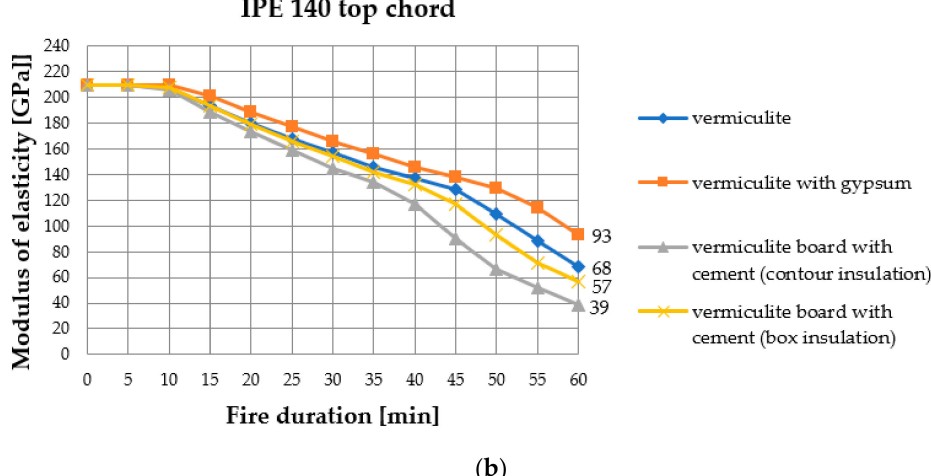

(**b**)

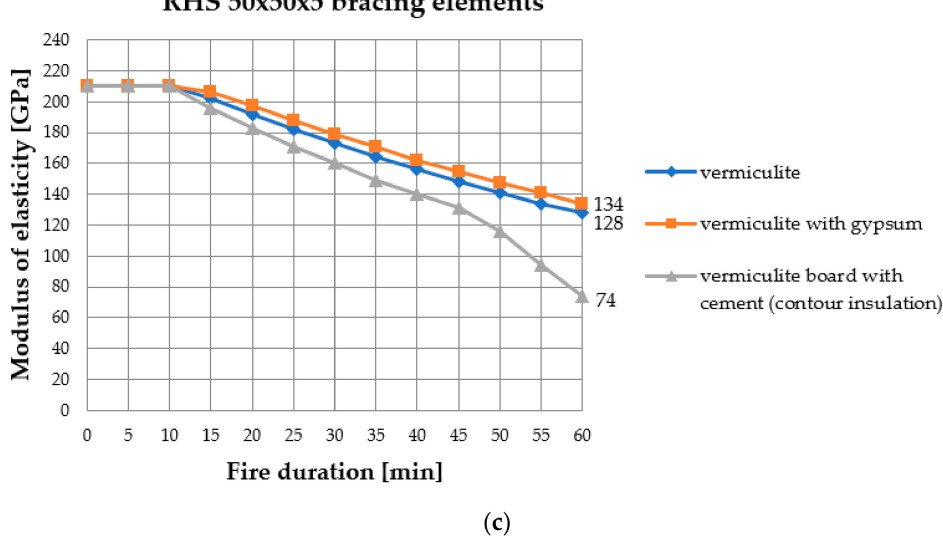

(**c**)

**Figure 4.** Decrease in modulus of elasticity under fire conditions (insulation 2 cm): (**a**) I 100 bottom chord, (**b**) IPE 140 top chord, (**c**) RHS 50X50X5 bracing elements.

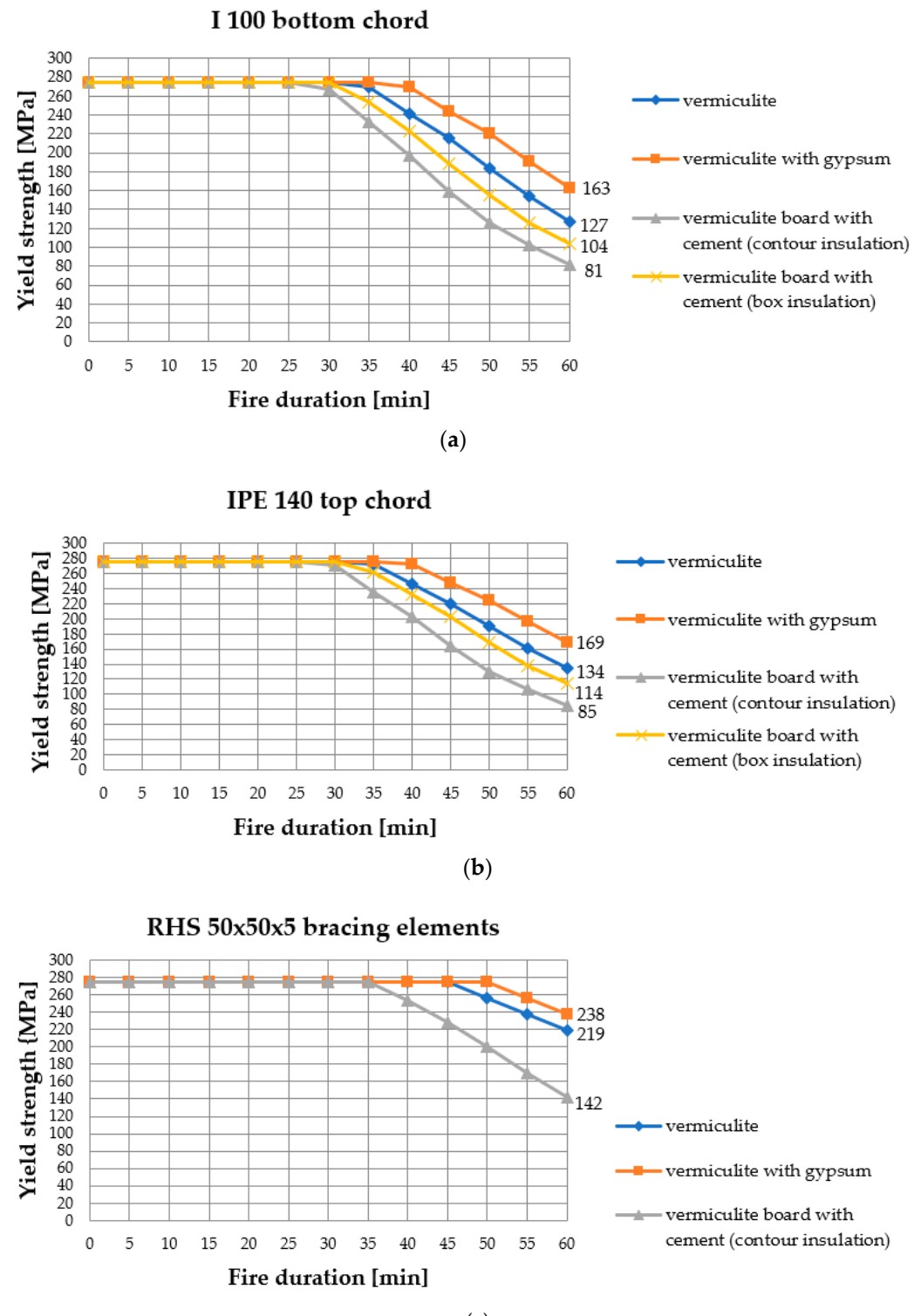

**Figure 5.** Decrease in yield strength under fire conditions (insulation 2 cm): (**a**) I 100 bottom chord, (**b**) IPE 140 top chord, (**c**) RHS 50X50X5 bracing elements.

The truss in Figure 1b is statically determined. A serial reliability system is suitable for such a structure. In this case, the failure of one of the most stressed elements is equivalent to a failure of the entire structure. Therefore, the resistance analysis includes the most stressed elements of the truss. Tension members of the bottom chord from number 2 to 5 have the same resistance. The most stressed of the compressed top chord were elements 9 and 10. Next, the resistances of compressed members

21,24 and tension members 20,25 were calculated. The analysis results are shown in Figure 6, in which the dotted line represents the effect of actions, i.e., internal forces arising in individual members due to the action of an external load.

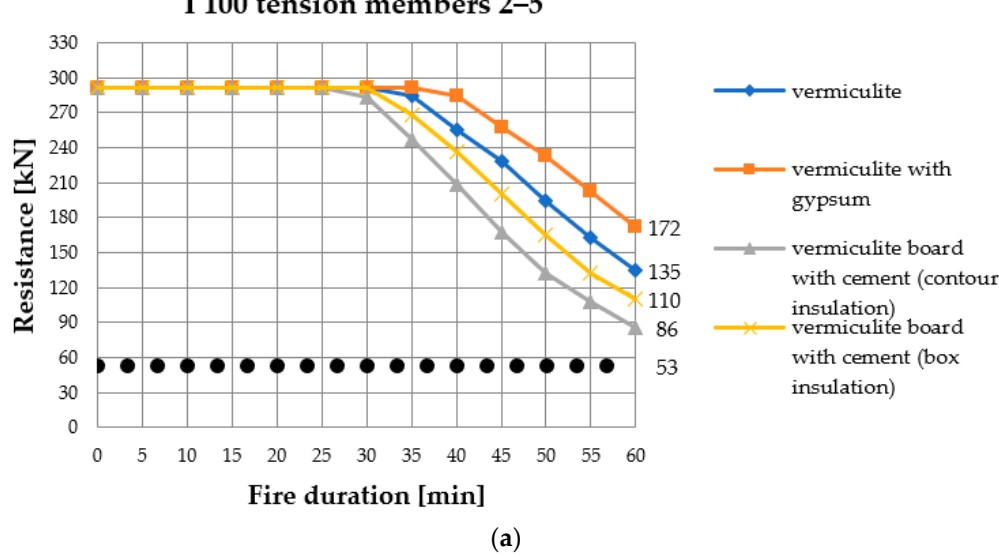

(**a**)

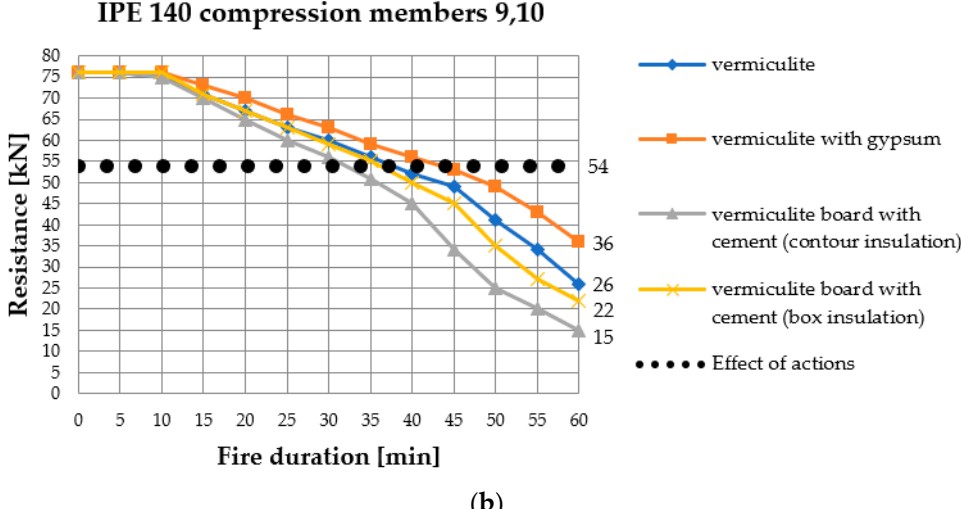

(**b**)

**Figure 6.** *Cont.*

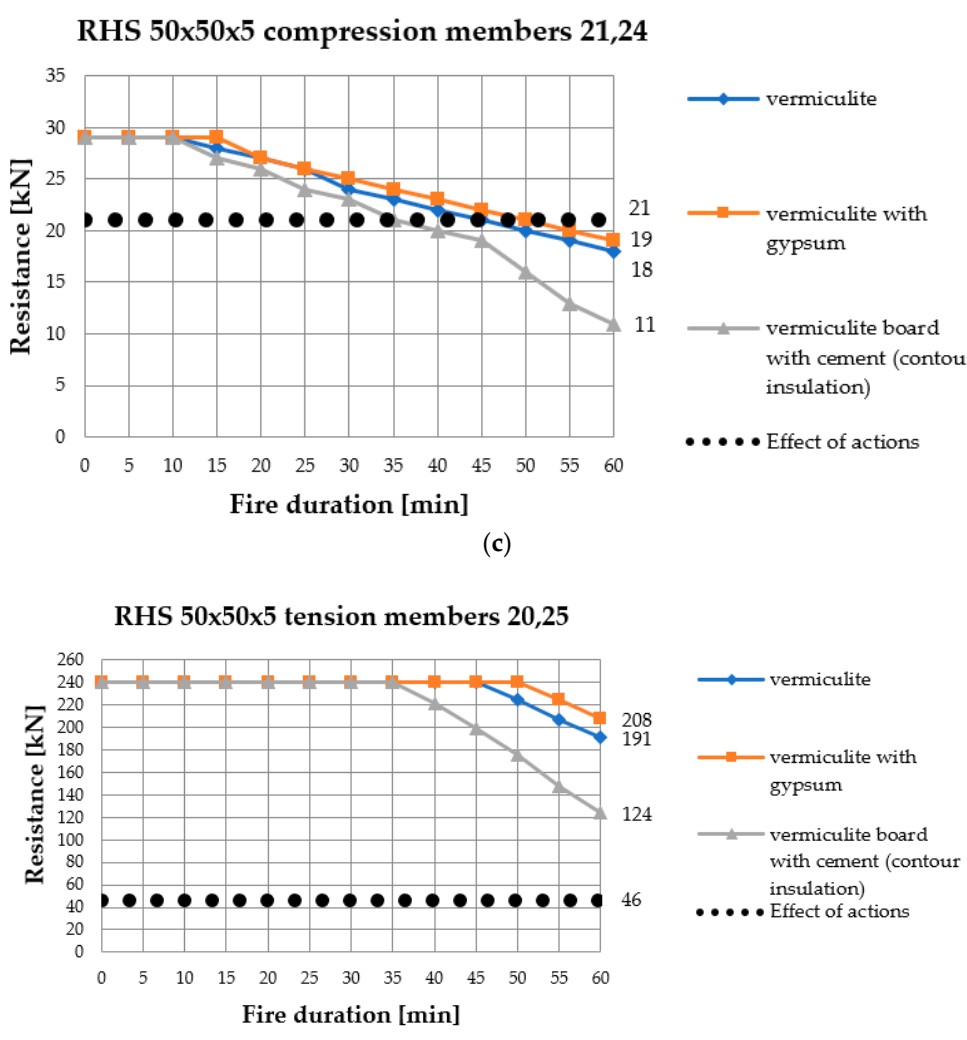

**Figure 6.** Resistance decrease in truss members under fire conditions (insulation 2 cm): (**a**) I 100 tension members 2–5, (**b**) IPE 140 compression members 9,10, (**c**) RHS 50X50X5 compression members 21,24, (**d**) RHS 50X50X5 tension members 20,25.

Analysing Figures 3–6, it is easy to notice that the best results were obtained by vermiculite with gypsum insulation. With this type of protection, steel members are heated up to the least extent value (Figure 3), which corresponds to a slower decrease in the modulus of elasticity (Figure 4) and yield strength (Figure 5) and, consequently, produces a milder reduction in resistance in the successive minutes of the fire (Figure 6). Slightly worse results were obtained using sprayed-on vermiculite insulation (Figures 3–6). In terms of fire analysis, this means density is one of the major parameters. The higher the density, the better fire-proofing properties a material has. Another important property is thermal conductivity; conversely, the material should have the lowest possible thermal conductivity value. In this analysis, the worst results were observed when the insulation from vermiculite and cement boards was applied. The boards had the highest density of all materials under consideration, but their thermal conductivity was over 50% higher relative to the sprayed-on insulation (Table 3). With vermiculite and cement boards, considerable improvements in results could be found using box insulation configuration instead of the contour one, which lowers the section factor. These results demonstrate that the lower the factor, the less heated up the element, which produces an advantageous effect on steel's mechanical properties in fire. It should also be emphasised that differences in the results

obtained for different types of insulation tend to grow as the fire develops. An adequate selection of fire-proofing insulation is of key importance for those members that require a high fire resistance class.

The diagrams in Figure 6 indicate that the use of 2 cm fire insulation makes it possible to reach the fire resistance class R30 for compression and class R60 for tension members of the truss. This result means that elements of bottom chord and tension members are able to withstand 60 min of exposure to fully developed fire, whereas compression elements are able to withstand only 30 min. The fire resistance class of the whole structures is equal to the lower fire resistance class of any single element. So, with the assumed insulation, the structure reached the R30 class.

The analysis of the graphs in Figure 6 clearly indicates a large capacity reserve in the tension elements. For instance, regarding the tension elements of the bottom chord, when 2 cm-thick vermiculite and gypsum insulation is used, the capacity of elements No. 2–5 after 60 min of fire duration is 172 kN, whereas the effect of action is 53 kN. That gives a reserve of 225%. A similar situation holds for the most stressed elements of the latticework, No. 20 and 25. After 60 min of the fire, their capacity is 208 kN while their effect of action amounts to 46 kN. Thus, the capacity reserve is 352%. Conversely, the performance of the compression elements is quite different. Figure 6 again illustrates this result. It can be seen that the most stressed elements of the top chords, No. 9 and 10, have a capacity of 36 kN and an effect of action of 54 kN. As a result, their capacity is exceeded by 33%. For the compression elements of the latticework, No. 21 and 24, their capacity exceedance is 9.5%.

On the basis of Figure 6, we can see that the commonly used method of constant insulation thickness is uneconomical. This paper proposes a differentiation of insulation thickness on truss elements. The best results were reached for the insulation of vermiculite with gypsum, so this material was used in the next analysis. The decrease of resistance during fire for different thicknesses of insulation is shown in Figure 7.

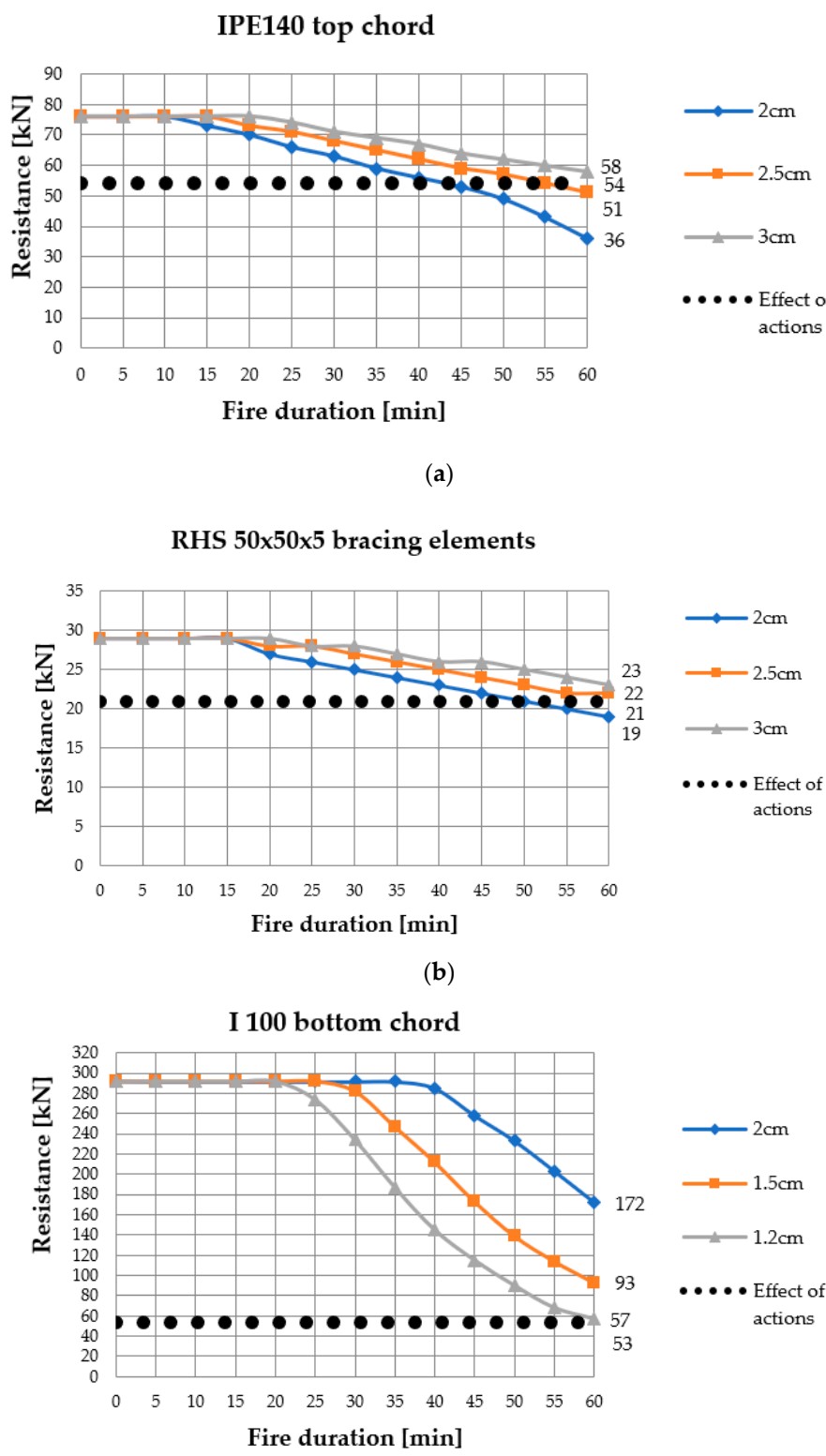

**Figure 7.** Resistance decrease in truss members under fire conditions depending on fire insulation thickness: (**a**) I 100 bottom chord, (**b**) RHS 50X50X5 bracing elements, (**c**) IPE 140 top chord.

According to charts in Figure 7, the thickness of fire insulation was altered. For the top chord it was increased to 3 cm; for bracing elements it was increased to 2.5 cm. The insulation thickness of the

tension elements of the bottom chord was decreased to minimum value of 1.5 cm (Table 3) to reduce the costs of fire protection. Finally, the structure reached the R60 fire resistance class.

## 4. Conclusions

In technical approvals, manufacturers of fireproofing materials usually provide tables in which the necessary insulation thicknesses are listed, depending on the type of material used, heating conditions, exposure factor, and the required fire resistance class. Those recommendations, however, do not account for the character of performance of individual structural members. The components in compression need thicker insulation than those in tension (Figure 6). This phenomenon is related to the fact that under fire conditions, the flexural buckling coefficient in compression members is abruptly reduced with an increase in temperature. In turn, this leads to a fast reduction in resistance (Figure 6b,c). In addition, members in tension have much higher resistance than those in compression in the basic design situation, i.e., at the instant of $t = 0$ min (Figure 6a,d). Consequently, even a considerable decrease in resistance of tension members is not as dangerous as that of compression members. The manufactures of fireproofing materials recommend that insulation of a constant thickness should be used for all elements of the truss. This approach seems highly uneconomical. In this paper, we propose an alternative solution, which determines the optimal necessary insulation thickness for individual bars of the truss based on additional static-strength calculations. The approach proposed in the paper, which is different from a conventional solution, is cost-effective in real terms. Our approach improves the ratio of fire safety to the cost of a fireproofing project.

**Author Contributions:** Introduction was prepared by K.K., U.P., and U.R. Definition of fire analysis and properties of insulating materials were written by K.K., U.P., and U.R. Results were obtained by K.K., U.P., and U.R. The analysis of the results and conclusions were written by K.K., U.P., and U.R.

**Funding:** Project financed under the programme of the Minister of Science and Higher Education under the name "Regional Initiative of Excellence" in the years 2019–2022; project number 025/RID/2018/19; amount of financing, 12 000 000 PLN.

**Conflicts of Interest:** The authors declare no conflict of interest.

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
