# Peer review of "Influence of the Thermal Insulation Type and Thickness on the Structure Mechanical Response Under Fire Conditions"

_applsci, doi:10.3390/app9132606_

Round 1

Reviewer 1 Report

The reviewed manuscript includes some interesting information. Unfortunatelly, the answered research question is unclear and it is difficult to understand the scope.

Furhermore, the English language of the manuscript needs improvement.

I believe the findings of the presented analysis could discussed in terms of "time delay coefficient".

Also discussion regarding the failure modes should be included.

In conclusion, I am not proposing the rejection of the paper, and I would like to give an opportunity to the authors to revise and improve.

Author Response

Urszula  Pawlak                                                                                      Kielce, 04.06.2019

Faculty of Civil Engineering and Architecture

Kielce University of Technology

Thank you for a careful review of our work.

In reference to critical and discussion comments and questions to authors included in the review of paper entitled “Influence of the thermal insulation type and thickness on the structure mechanical response under fire conditions”, I post below relevant explanation.

Responses to specific comments

1.       The reviewed manuscript includes some interesting information. Unfortunately, the answered research question is unclear and it is difficult to understand the scope.

We agree completely with the opinion of the reviewer. In the new, improved version of the article many changes are introduced. In authors’ opinion the text is now much more clear and understandable. Also the innovation of presented methodology is underlined.

2.      Furthermore, the English language of the manuscript needs improvement.

English language was improved.

I believe the      findings of the presented analysis could discussed in terms of "time      delay coefficient".

 The results of the thermal analysis carried out at work could be described by the terms of "time delay coefficient". This is a very interesting approach. We got acquainted with the work of Zhi-Hua Wang Kang Hai Tan (Fire Safety Journal, Volume 41, Issue 1, February 2006, Pages 31-38) in which it was conducted sensitivity study of time delay coefficient of heat transfer formulations for insulated steel members exposed to fire. We intend to develop this approach in further research. We performed the thermal analysis in accordance with the current Eurocode. The main purpose of our work was to obtain a mechanical response of the structure in terms of the effect of actions and resistance.

In technical approvals, manufacturers of fire-proofing materials do not account for the character of performance of individual structure members. The components in compression need thicker insulation than those in tension. That is related to the fact that under fire conditions, the flexural buckling coefficient in compressed members is abruptly reduced with an increase in temperature. In turn, that leads to a fast reduction in resistance. In addition, members in tension have much higher resistance than those in compression in the basic design situation, i.e. at the instant of t=0min. Consequently, even a considerable decrease in resistance of tension members is not as dangerous as that of compression members. Therefore, due to the nature of the performance of individual elements, fire-proofing insulation of every steel structure should be computationally verified. The manufactures of fireproofing materials recommend that insulation of a constant thickness should be used for all elements of the truss. This approach seems highly uneconomical. The authors of the paper have proposed an alternative solution. The advantage of the proposed solution is a more optimal (economical) method of fire insulation of steel lattice girders. Additionally in the paper, the influence of the type of fire insulation on the mechanical response of the structure was investigated. Calculations were carried out for different types of sprayed-on insulation, and also for contour and box insulation panels. The graphs show the behaviour of the elastic modulus, the yield point and the resistance of the elements in the successive minutes of the fire for different means of fire protection used. The best results were obtained for vermiculite and gypsum spray.

4.      Also discussion regarding the failure modes should be included.

The truss in Figure 1 is statically determined. A serial reliability system is suitable for such
a structure. In this case, the failure of one of the most stressed elements is equivalent to failure of the entire structure. Therefore, the resistance analysis included the most stressed elements of the truss. Tension members of the bottom chord from number 2 to 5 have the same resistance. As regards the compressed top chord the most stressed were  elements 9 and 10. Next the resistances of compressed members 21,24 and tension members 20,25 were calculated. The analysis results are shown in Figure 6, in which the dotted line represents the effect of actions, i.e. internal forces arising in individual members due to the action of external load.

Reviewer 2 Report

Thanks for inviting me to review the manuscript entitled "Influence of the thermal insulation type and thickness on the structure mechanical response under fire conditions".

This manuscript report a research on the effect of thermal insulation type and thickness on the the load-carrying capacity of a simple steel truss exposed to elevated temperature.  While the research topic - structural fire safety - is important, this research does not provide sufficient technical contributions, so I cannot recommend it for publication before it is significantly improved.

Following are some detailed comments that might be useful for improvement.

(1) In your introduction, please provide more information to clarify the novelty and technical contributions of this research.

(2) Your methodology is unclear. It is insufficient to just mention "MES3D program". Did you perform finite element analysis? If yes, please provide more details of your finite element model. 

(3) What does the analyzed truss stand for? Is it for a bridge or a roof of building?

(4) Please elaborate how you applied the insulation material to the truss?

(5) How did you validate your method and analysis results?

(6) Please elaborate how you evaluated the degradation of the load-carrying capacity. There are many recent studies on structural fire safety evaluation and assessment you may consider to refer. For instance: Li et al. (2019). Post-fire seismic behavior of two-bay two-story frames with high-performance fiber-reinforced cementitious composite joints. Engineering Structures, 183, pp.150-159.

(7) The conclusions should be rewritten. Please clearly point out the new findings from this research.

(8) The language should be improved, and the format of figures should be consistent.

Author Response

Urszula  Pawlak                                                                                      Kielce, 04.06.2019

Faculty of Civil Engineering and Architecture

Kielce University of Technology

Thank you for a careful review of our work.

In reference to critical and discussion comments and questions to authors included in the review of paper entitled “Influence of the thermal insulation type and thickness on the structure mechanical response under fire conditions”, I post below relevant explanation.

Responses to specific comments

1.      In your introduction, please provide more information to clarify the novelty and technical contributions of this research.

In the new, improved version of the article many changes are introduced. In authors’ opinion the text is now much more clear and understandable. Also the innovation of presented methodology is underlined.

Your methodology is unclear.      It is insufficient to just mention "MES3D program". Did you      perform finite element analysis? If yes, please provide more details of      your finite element model. 

We agree completely with the opinion of the reviewer that information about program MES3D are insufficient. It is corrected (lines 231-266 of the chapter 3. Result and discussion).

What does the analyzed truss      stand for? Is it for a bridge or a roof of building?

The truss shown in Figure 1 is the element of the load-bearing hall with lattice girders based on prefabricated reinforced concrete pillars.

 Please elaborate how you      applied the insulation material to the truss?

The method applied the insulation material to the truss is elaborated in line 280-312, chapter 3. Result and discussion.

How did you validate your      method and analysis results?

The research and their results carried out in the article were not verified by experiment. Non-destructive testing is more expensive, and at the moment we do not have adequate finances.

Please elaborate how you      evaluated the degradation of the load-carrying capacity. There are many      recent studies on structural fire safety evaluation and assessment you may      consider to refer. For instance: Li et al. (2019). Post-fire seismic      behavior of two-bay two-story frames with high-performance fiber-reinforced      cementitious composite joints. Engineering      Structures, 183,
     pp.150-159.

The approach proposed in Eurocode [22] was applied.

7.      The conclusions should be rewritten. Please clearly point out the new findings from this research.

We agree completely with the opinion of the reviewer, the conclusions were rewritten.

8.      The language should be improved, and the format of figures should be consistent.

English language and the format of figures were improved.

Round 2

Reviewer 1 Report

The paper has been improved. I believe some discussion can be added regarding the failure modes and fire resistance critiria. Possibly, useful literature can be (but not limited to)

Wang,  Y 2018, Composite Structures of Steel and Concrete: Fire  Resistance. in R Johnson (ed.), Composite Structures of Steel and Concrete. 4 edn, John Wiley & Sons Ltd,               Chichester, pp. 223-245.

Ozyurt, E & Wang, Y 2016, 'Effects of Non-Uniform Temperature Distribution on Critical Member Temperature of Steel Tubular Truss', Engineering Structures, vol. 116, pp.      95-106. https://doi.org/10.1016/j.engstruct.2016.02.044

Designing Steel Structures for Fire Safety 1st Edition Jean Marc Franssen, Venkatesh Kodur, Raul Zaharia, CRC Press

Maraveas, C., Local buckling of steel members under fire conditions: A review, Fire Technology, Volume 55, Issue 1, pp 51–80, 2019 https://doi.org/10.1007/s10694-018-0768-1

Maraveas, C., Fire resistance of DELTABEAM® composite beams: a numerical investigation, Journal of Structural Fire Engineering, Vol. 8 Issue: 4, pp. 338-353, 2017 https://doi.org/10.1108/JSFE-05-2016-0003

Maraveas, C., Tsavdaridis, K.D., Nadjai, A., Fire resistance of unprotected Ultra Shallow Floor Beams: A numerical investigation, Fire Technology, Volume 53, Issue 2, pp 609–627, 2017, DOI: 10.1007/s10694-016-0563-9

Author Response

Urszula  Pawlak                                                                                      Kielce, 11.06.2019

Faculty of Civil Engineering and Architecture

Kielce University of Technology

Thank you for your time and valuable tips to improve our work

Responses to specific comments

The paper has been improved. I believe some      discussion can be added regarding the failure modes and fire resistance      critiria. Possibly, useful literature can be (but not limited to).

[1] Wang,  Y 2018, Composite Structures of Steel and Concrete: Fire  Resistance. in R       Johnson (ed.), Composite Structures of Steel and Concrete. 4 edn, John Wiley &       Sons Ltd, Chichester, pp. 223-245.

[2] Ozyurt, E & Wang, Y 2016, 'Effects of Non-Uniform Temperature Distribution on       Critical Member Temperature of Steel Tubular Truss', Engineering Structures, vol.       116, pp. 95-106. https://doi.org/10.1016/j.engstruct.2016.02.044

[3] Designing Steel Structures for Fire Safety 1st Edition Jean Marc Franssen,       Venkatesh Kodur, Raul Zaharia, CRC Press

[4] Maraveas, C., Local buckling of steel members under fire conditions: A        review, Fire Technology, Volume 55, Issue 1, pp 51–80, 2019        https://doi.org/10.1007/s10694-018-0768-1

[5]  Maraveas, C., Fire resistance of DELTABEAM® composite beams: a numerical        investigation, Journal of Structural Fire Engineering, Vol. 8 Issue: 4, pp. 338-        353, 2017 https://doi.org/10.1108/JSFE-05-2016-0003

[6]  Maraveas, C., Tsavdaridis, K.D., Nadjai, A., Fire resistance of unprotected Ultra        Shallow Floor Beams: A numerical investigation, Fire Technology, Volume        53, Issue 2, pp 609–627, 2017, DOI: 10.1007/s10694-016-0563-9

The authors familiarized themselves with the mentioned paper [1,2,3,4,5,6].

The use of cold formed structural elements has become more common in the construction industry during the past decades due to the various advantages it offers, such as weight reduction of the structural system. The thin-walled sections differ from hot-rolled sections in terms of failure modes. In addition to local and global buckling, distortional buckling is also possible. The slenderness of steel members is a major factor in determining their failure mode at elevated temperatures. Short, stocky elements typically have a slenderness ratio less than 40, while in long, slender elements this value is greater than 120. The stocky columns exhibited local buckling as their failure mode, while longer, more slender columns displayed global buckling accompanied by limited local buckling. Problems related to the description of failure modes were the subject of discussion in many works. Articles deserve special attention [2,3,4]

Reviewer 2 Report

The quality of the revised manuscript has been improved, and many of my concerns have been addressed. However, some key information is still missing or unclear. Further improvement should be made. 

Here are some detailed comments and questions:

(1) More detailed information of the finite element model must be provided. For instance:

How did you establish the model? What elements did you use? What were the boundary conditions? How did you apply the loading?

(2) How did you consider the temperature distribution of the analyzed structure? Did you assume uniform temperature at different positions? In recent research, distributed fiber optic sensors have been used to measure the temperature distributions in beams exposed to fire. It was found that the temperature distribution is nonuniform. See: Bao et al. (2019). "Review of fiber optic sensors for structural fire engineering", Sensors, 19(4), 877.

(3) I do not know the importance of the analyzed 2D truss structure. Are the conclusions generated from this study useful for real structures? Please justify. 

(4) This study is based the finite element model, so the model must be verified. Otherwise, I am not convinced by the results and conclusions. 

(5) Please comment on the role of high-performance fiber-reinforced cementitious composites (HPFRCC) in fire safety of buildings. Recently, HPFRCC has been developed and applied to improve the fire resistance of RC columns and RC beam-column frames in buildings.  

(6) The references are not properly formatted and cited. Please double check and correct. 

Author Response

Urszula  Pawlak                                                                                      Kielce, 11.06.2019

Faculty of Civil Engineering and Architecture

Kielce University of Technology

Thank you for your time and valuable tips to improve our work

Responses to specific comments

More detailed information of the finite element      model must be provided. For instance: How did you establish the model?      What elements did you use? What were the boundary conditions? How did you      apply the loading?

The structure shown in Figure 1a is a steel, industrial hall with lattice girders pinned jointed to prefabricated reinforced concrete columns. The bracing system commonly used in this buildings makes it possible to model lattice girds as a flat trusses (Figure 1b). This approach is widely used in the design practice. The connection  of a steel lattice girder and a column is implemented as pinned.  That's why it was assumed that the truss is simply supported. The system was loaded only with concentrated forces applied to the nodes of the top chord. In Figure 1b, the compression elements are marked in red and the tension members in green.

Figure 1a Example of an industrial hall with steel lattice girders [http://www2.um.bydgoszcz.pl/city/news/News-March-2015 /Fair center_ Columns _ ready_ Report_nr_3.aspx?option=print]

Figure 1b. Static scheme of the flat truss model

In the static analysis by means FEM we used a flat truss element. It is a simple two-node element with two degrees of freedom in each node.` The shape functions of the displacement field are linear.  The displacement field of an element contains two translational components: horizontal u and vertical v. The strain tensor is reduced to one non-zero component of Green’s strain tensor ε11 which characterizes the elongation of the bar. Other components of the strains tensor are equal to zero. The stress tensor is represented by component σ11 of Pioli-Kirchhoff’s second stress tensor. The detailed mathematical formulations of the stress-strain relationship for steel at elevated temperatures are given in [23]. In Figure 2 this relationship is shown for steel S275.

Figure 2.  Stress-strain relationship for steel S275

The starting point of a full fire analysis is to define fire scenarios. It is assumed that the structure is in the fully developed fire phase, consequently the standard fire curve Eqs.(1) is adopted. For unprotected and insulated steel members, temperature increment could be estimated in accordance with the formulas Eqs.(2) and Eqs. (3). The temperature field is uniform along the member length and along the cross-section height. Next, the mechanical properties that influence the resistance of tension and compression members are calculated. The most important mechanical properties of steel elements are yield strength and Young’s modulus of elasticity. A decrease in the values of mechanical parameters leads directly to a reduction in the resistance of tension and compression members Eqs. (5), Eqs.(6), which finally results in the limit state being exceeded. Thermal load is brought into the nodes as an additional load, in accordance with the incremental FEM method. For thermal load defined in this way, the stiffness matrix of the truss element is determined for Young’s modulus
as a function of temperature.

As regards fire analysis, the authors did not intend to determine the value of the limit load, but that of the fire resistance, i.e. to specify the time of the structure failure. In this particular case, static loads are constant, whereas additional axial forces can be created in the truss elements under thermal load. Due to the fact that the structure is statically determinate, during the whole fire analysis, internal forces remain the same as in the basic situation. The computations, aimed at finding axial forces, strains and displacements, are run with MES3D program [24,25,26]. The computations proceeded in two steps. The first step (200C) produced the specification of the state of the structure in a basic design situation. In the second step, temperature increments related directly to time steps, are taken into account. The analysis was repeated every second.

2.      How did you consider the temperature distribution of the analyzed structure? Did you assume uniform temperature at different positions? In recent research, distributed fiber optic sensors have been used to measure the temperature distributions in beams exposed to fire. It was found that the temperature distribution is nonuniform. See: Bao et al. (2019). "Review of fiber optic sensors for structural fire engineering", Sensors, 19(4), 877.

The temperature field is uniform along the member length and along the cross-section height. The authors familiarized themselves with the mentioned paper [1]. In the future we are going to improved our thermal model. Next step of work will be temperature distribution as nonuniform

[1] Yi Bao, Ying Huang , Matthew S. Hoehler, Genda Chen, Review of Fiber Optic Sensors for Structural Fire Engineering, Sensors 201919(4), 877; https://doi.org/10.3390/s19040877

3.      I do not know the importance of the analyzed 2D truss structure. Are the conclusions generated from this study useful for real structures? Please justify.

The structure shown in Figure 1a is a steel, industrial hall with lattice girders pinned jointed to prefabricated reinforced concrete columns. The bracing system commonly used in this buildings makes it possible to model lattice girds as flat trusses (Figure 1b). This approach is widely used in the design practice.  The connection  of a steel lattice girder and a column is implemented as pinned.  That's why it was assumed that the truss is simply supported.

4.      This study is based the finite element model, so the model must be verified. Otherwise, I am not convinced by the results and conclusions.

The fire behaviour of bearing elements of engineering structures can be assessed on the basis of experimental research and / or finite element analysis[1]. Currently, design techniques are based mainly on the analysis of the fire phenomenon by means of FEM. In the work, calculations were carried out in parallel using two programs: author's MES 3D and commercial Autodesk Robot, and the same results were obtained. In our case, the verification of the model was based only on numerical simulation. Conducting experimental research was our main goal, which we will accomplish in the next research step, after obtaining funds in 2020. It will be possible thanks to cooperation with the industry.

[1]. XiulingLi, ZhenboXu,  YiBao, ZhengangCong, Post-fire seismic behaviour of two-bay two-story frames with high-performance fiber-reinforced cementitious composite joints, Engineering Structures, 2019, Volume 183, 150-159,

5.      Please comment on the role of high-performance fiber-reinforced cementitious composites (HPFRCC) in fire safety of buildings. Recently, HPFRCC has been developed and applied to improve the fire resistance of RC columns and RC beam-column frames in buildings. 

The HPFRCC high performance fiber-reinforced cementitious composite is a mixture of the right proportions of Portland cement, fly ash, quartz sand, water and polyvinyl alcohol (PVA) fibers. It is PVA fibers that have a decisive influence on the improvement of mechanical efficiency, elasticity and durability of concrete structures [1,2] due to fire conditions. According to Li, Xu, Bao, Cong [3], PVA fibers under the influence of high temperatures melt to form channels to alleviate internal vapor pressure and thus explosive spalling. Therefore, the mechanical properties after a fire are higher compared to conventional concrete. In [3], the authors performed experimental studies of four two-story and two-nave frames, two made of typical concrete (S1 and S2) and two, in which the beam-column joint was made of high-performance cement composite reinforced with HPFRCC fibers (S3 and S4). The frames S2, S3, S4 were subjected to fire. The results of the tests carried out in the fire chambers showed that the application of HPFRCC increased the initial stiffness by 30%, resistance and ultimate displacement by 6% and 3%, the energy dissipation by 21%. In additionally only small surface flakes appeared at the HPFRCC joints, without the participation of explosive spalling on the HPFRCC.

[1] Meng W, Valipour M, Khayat KH. Optimization and performance of cost-effective ultra-high performance concrete. Mater Struct 2017;50(1):29.

[2] Meng W, Khayat KH. Effects of saturated lightweight sand content on key characteristics of ultra-high-performance concrete. Cem Concr Res 2017;101:46–54.

[3] XiulingLi, ZhenboXu,  YiBao, ZhengangCong, Post-fire seismic behaviour of two-bay two-story frames with high-performance fiber-reinforced cementitious composite joints, Engineering Structures, 2019, Volume 183, 150-159,

6.      The references are not properly formatted and cited. Please double check and correct.

The references were improved.

Round 3

Reviewer 2 Report

The authors have addressed my comments. 

The quality of figures should be significantly improved before it is published. 

As pointed out, the format of references must be carefully checked and corrected. 

Author Response

Urszula  Pawlak                                                                                      Kielce, 21.06.2019

Faculty of Civil Engineering and Architecture

Kielce University of Technology

Thank you for your time and valuable tips to improve our work

Responses to specific comments

The quality of      figures should be significantly improved before it is published.

The quality of figures was improved.

2.      As pointed out, the format of references must be carefully checked and corrected.

The format of references was carefully checked and corrected.
